# Short- and Long-Term Efficacy and Safety of Deep-Brain Stimulation in Parkinson’s Disease Patients aged 75 Years and Older

**DOI:** 10.3390/brainsci12111588

**Published:** 2022-11-20

**Authors:** Chao Jiang, Jian Wang, Tong Chen, Xuemei Li, Zhiqiang Cui

**Affiliations:** 1Institute of Neuroscience, College of Life and Health Sciences, Northeastern University, Shenyang 110169, China; 2Department of Neurosurgery, The First Medical Center of Chinese PLA General Hospital, Beijing 100853, China; 3Department of Neurology, The Second Medical Center & National Clinical Research Center for Geriatric Diseases, Chinese PLA General Hospital, Beijing 100853, China; 4Cadre Medical Department, The First Medical Clinical Center, PLA General Hospital, 28 Fuxing Road, Haidian District, Beijing 100853, China

**Keywords:** older patients, Parkinson’s disease, deep brain stimulation, levodopa equivalent daily dose (LEDD)

## Abstract

Objective: The aim of this study was to investigate the efficacy and safety of deep-brain stimulation (DBS) in the treatment of patients with Parkinson’s disease aged 75 years and older. Methods: From March 2013 to June 2021, 27 patients with Parkinson’s disease (≥75 years old) who underwent DBS surgery at the First Medical Center of the PLA General Hospital were selected. The Unified Parkinson’s Disease Rating Scale Part 3 (UPDRS-III), 39-item Parkinson’s Disease Questionnaire (PDQ-39), and Barthel Index for Activities of Daily Living (BI) scores were used to evaluate motor function and quality of life before surgery and during on and off periods of DBS at 1 year post operation and at the final follow-up. A series of non-motor scales were used to evaluate sleep, cognition, and mood, and the levodopa equivalent daily dose (LEDD) was also assessed. Adverse events related to surgery were noted. Results: The average follow-up time was 55.08 (21–108) months. Symptoms were significantly improved at 1 year post operation. The median UPDRS-III score decreased from 35 points (baseline) to 19 points (improvement of 45.7%) in the stimulation-on period at 1 year post operation (t = 19.230, *p* < 0.001) and to 32 points (improvement of 8.6%) at the final follow-up (t = 3.456, *p* = 0.002). In the stimulation-off period, the median score of UPDRS-III increased from 35 points to 39 points (deterioration of −11.4%) at 1 year post operation (Z = −4.030, *p* < 0.001) and 45 points (deterioration of −28.6%) at the final follow-up (Z = −4.207, *p* < 0.001). The PDQ-39 overall scores decreased from 88 points (baseline) to 55 points (improvement of 37.5%) in the stimulation-on period at 1 year post operation (t = 11.390, *p* < 0.001) and 81 points (improvement of 8.0%) at the final follow-up (t = 2.142, *p* = 0.044). In the stimulation-off period, the median PDQ-39 score increased from 88 points to 99 points (deterioration of −12.5%) at the final follow-up (Z = −2.801, *p* = 0.005). The ADL-Barthel Index score increased from 25 points (baseline) to 75 points (improvement of 66.7%) at 1 year post operation (Z = −4.205, *p* < 0.001) and to 35 points (improvement of 28.6%) at the final follow-up (Z = −4.034, *p* < 0.001). In the stimulation-off period, BI scores decreased from 25 points to 15 points (deterioration of −40%) at 1 year post operation (Z = −3.225, *p* = 0.01) and to 15 points (deterioration of −40%) at the final follow-up (Z = −3.959, *p* = 0.001). Sleep, cognition, and mood were slightly improved at 1 year post operation (*p* < 0.05), and LEDD was reduced from 650 mg (baseline) to 280 mg and 325 mg at 1 year post operation and the final follow-up, respectively (*p* < 0.05). One patient had a cortical hemorrhage in the puncture tract on day 2 after surgery, five patients had hallucinations in the acute stage after surgery, and one patient had an exposed left-brain electrode lead at 4 months post operation; there were no infections or death. Conclusion: DBS showed efficacy and safety in treating older patients (≥75 years old) with Parkinson’s disease. Motor function, quality of life, activities of daily living, LEDD, and sleep all showed long-term improvements with DBS; short-term improvements in emotional and cognitive function were also noted.

## 1. Introduction

For Parkinson’s disease (PD) patients whose symptoms cannot be controlled by drugs, deep-brain stimulation (DBS) is an effective treatment option [1,2,3,4]. DBS can provide significant control of motor symptoms, levodopa-induced dyskinesias, and a substantial improvement in quality of life (QoL) for PD patients [5,6,7,8,9]. Despite the experience that has been gained over time with DBS therapy, the age limit for the procedure remains a matter of debate. Because of increased surgical complications and reduced surgical benefits, the number of DBS operations in older PD patients remains relatively low. The same applies to clinical trial limits; some studies considered 75 years old as the cutoff [8,10], others considered 80 years [11], and others had no specified maximum age [1,2,4]. In recent years, although there have been reports of DBS in PD patients over 70 years old [12], even in those aged 75 years and over [13,14], there are still relatively few such reports. With age, many brain functions decline, including working and episodic memory, cognitive and emotional function, decision making, and executive function [15]. These age-related changes can affect the efficacy and safety of DBS surgery; hence, age is an important element for consideration. In the present study, we report both the short- and the long-term efficacy of DBS in older (≥75 years old) PD patients, and we also analyze changes in emotion, sleep, cognitive function, levodopa equivalent daily dose (LEDD), and complications after DBS surgery.

### 1.1. Materials and Methods

#### 1.1.1. Patients

From March 2013 to June 2020, 27 elderly PD patients (≥75 years old) received DBS in the Neurosurgery Department of the First Medical Center of the Chinese People’s Liberation Army General Hospital. Of these patients, 23 were followed up. Four patients had their follow-up interrupted; one (Patient 24) was unable to be contacted after going abroad for recuperation, one (Patient 25) had their follow-up interrupted by family members who did not cooperate, one (Patient 26) had a cerebral infarction and died after 6 months of bed rest, and one (Patient 27) died of multiple organ failure caused by wrestling after the operation.

We extracted relevant data at 1 week before surgery (off-medication state), 1 year after surgery (off-medication/on-stimulation state), and the final follow-up (off-medication/on-stimulation state). The indicators of interest were as follows: (1) evaluation of motor symptoms and QoL using the Unified PD Rating Scale part 3 (UPDRS-III) [16], 39-item PD Questionnaire (PDQ-39) [17], and Barthel Index for Activities of Daily Living (BI) [18]; (2) assessment of sleep using the PD Sleep Scale—Chinese Version (PDSS-CV) [19], Pittsburgh Sleep Quality Index (PSQI) [20], and Epworth Sleepiness Scale (ESS) [21]; (3) evaluation of cognitive function using the Montreal Cognitive Assessment (MoCA) [22] and Mini-Mental State Examination (MMSE) [23]; (4) assessment of emotional states (anxiety and depression) using the Hamilton Anxiety Scale (HAMA) [24] and Hamilton Depression Scale (HAMD) [25]; (5) the average levodopa equivalent daily dose (LEDD) [26]; (6) any intraoperative or postoperative adverse events.

#### 1.1.2. Inclusion and Exclusion Criteria

All patients voluntarily participated in this study and signed an informed consent form. For the inclusion and exclusion criteria for surgery, we referred to the previous literature [13,27,28,29,30]. The following inclusion criteria were used for eligible participants: (1) a 4 year diagnosis of idiopathic PD [28]; (2) a levodopa/apomorphine challenge test demonstrating >30% improvement in UPDRS-III score [29]; (3) motor fluctuations and/or severe dyskinesia; (4) QoL and social functioning influenced by levodopa-responsive signs; (5) clinical manifestations in off periods (unpredictable off periods, severe off periods (pain, dystonia, or panic attack), and off time more than 25% in the daytime) [13].

Because patients were relatively old, we used strict exclusion criteria to minimize complications, as follows: (1) high surgical risk (e.g., severe atrophy or diffuse vascular white-matter changes in brain magnetic resonance imaging (MRI)) [13]; (2) severe levodopa-resistant axial impairment (balance, speech, or gait problems) [30]; (3) suicide risk, moderate or severe depression (HAMD ≥ 17), or severe anxiety (HAMA ≥ 24); (4) moderate or severe cognitive impairment or mild cognitive impairment with progressive features (MMSE < 27 or MoCA < 26); (5) a levodopa/apomorphine challenge test demonstrating ≤30% improvement in UPDRS-III score.

#### 1.1.3. Surgical Methods

All patients underwent head MRI scans (1.5 or 3.0 T scanner; MAGNETOM Espree; Siemens Healthineers, Erlangen, Germany) 1 to 2 days before surgery, and the operation plan was developed. A computed tomography CT scan was performed after the Leksell head frame (Leksell; Elekta, Stockholm, Sweden) was installed on the day of surgery; the two sets of data were fused to determine the frame coordinates. One of the patients was unable to undergo an MRI scan because of a cardiac stent; a preoperative CT scan was used to formulate the surgical plan. Given that all patients were of advanced age, levodopa drugs were discontinued before surgery. Because the symptoms of drug withdrawal were obvious, surgery was performed under general anesthesia; microelectrode recording (two channels) was performed during the operation. Except for one patient with cardiac stents, intraoperative MRI scans (Siemens Espree, 1.5 T, Siemens Healthineers) were performed to exclude complications such as intraoperative bleeding or ischemia and to confirm the accuracy of electrode positioning. Lastly, a pulse generator was placed in the patient’s chest. The surgical procedure was described in detail in our previous studies [31,32,33]. Nineteen patients underwent bilateral subthalamic nucleus (STN)-DBS, one patient with Parkinson/dystonia syndrome underwent bilateral globus pallidus internus (Gpi)-DBS, one patient with tremor as the main manifestation underwent bilateral thalamic ventral intermediate nucleus (Vin)-DBS, and one patient underwent left Vin-DBS and right STN-DBS.

#### 1.1.4. Statistical Analysis

Statistical analysis was performed using SPSS version 24.0 (IBM, Armonk, NY, USA). Measurement data that conformed to a normal distribution were expressed as the mean ± standard deviation, those that did not conform were expressed as the median (75th percentile–25th percentile), and numerical data were expressed as rates and percentages. The Wilcoxon signed-rank test was used to analyze the data, Pearson correlation analysis was used to analyze correlations, and *p* < 0.05 was considered significant.
$$\mathrm{Improvement}\;\mathrm{rate}\;(\%)=\frac{\left|\mathrm{postoperative}\;\mathrm{score}-\;\mathrm{preoperative}\;\mathrm{score}\right|}{\mathrm{preoperative}\;\mathrm{score}}\times 100\%.$$

## 2. Results

### 2.1. Clinical Characteristics of DBS in Elderly PD Patients

The effective rate of return visit was 85.2%, including 16 males and seven females; the mean age of patients was 77.0 (75–85) years. The shortest medical history was 60 months, and the longest was 432 months, with an average medical history of 119.9 months. Follow-up time ranged from 21 to 108 months, with an average of 55.1 months. There were seven patients (30.4%; Patients 7, 11, 12, 13, 19, 22, and 23) with essential hypertension, four patients (17.4%; Patients 3, 5, 10, and 15) with thyroidectomy, four patients (17.4%; Patients 2, 5, 8, and 19) with cholecystectomy, three patients (13.0%; Patients 8, 12, and 14) with diabetes, and three patients (13.0%; Patients 1, 3, and 12) with hallucinations (Table 1).

### 2.2. Postoperative Efficacy of DBS in Elderly PD Patients

The average follow-up time was 55.08 (21–108) months. Overall, conditions were significantly improved at 1 year post operation (Figure 1, Table 2). The UPDRS-III scores decreased from 35 points (baseline) to 19 points at 1 year post operation (stimulation-on period; t = 19.230, *p* < 0.001), and to 32 points at the final follow-up (stimulation-on period; t = 3.456, *p* = 0.002). This corresponds to an improvement of 45.7% at 1 year and 8.6% at the last follow-up. In the stimulation-off period, median UPDRS-III scores increased from 35 points to 39 points 1 year post operation (Z = −4.030, *p* < 0.001), and 45 points at the final follow-up (Z = −4.207, *p* < 0.001), which corresponds to a deterioration of −11.4% at 1 year and −28.6% at the last follow-up. In terms of patients’ QoL, overall PDQ-39 scores decreased from 88 points (baseline) to 55 points at 1 year post operation (stimulation-on period; t = 11.390, *p* < 0.001), and 81 points at the final follow-up (stimulation-on period; t = 2.142, *p* = 0.044). This corresponds to an improvement of 37.5% at 1 year and 8.0% at the last follow-up. In the stimulation-off period, median PDQ-39 scores increased from 88 points to 99 points at the final follow-up (Z = −2.801, *p* = 0.005), which corresponds to a deterioration of −12.5% at the last follow-up. The ADL-Barthel Index score increased from 25 points (baseline) to 75 points at 1 year post operation (stimulation-on period; Z = −4.205, *p* < 0.001), and to 35 points at the final follow-up (stimulation-on period; Z = −4.034, *p* < 0.001). This corresponds to an improvement of 66.7% at 1 year and 28.6% at the last follow-up. In the stimulation-off period, scores decreased from 25 points to 15 points at 1 year post operation (Z = −3.225, *p* =0.01), and to 15 points at the final follow-up (Z = −3.959, *p* = 0.001), which corresponds to a deterioration of −40% at 1 year and −40% at the last follow-up.

In terms of sleep, compared with baseline scores, PDSS-CV scores at both 1 year post operation and the final follow-up were improved (Z = −3.669, *p* < 0.001 and Z = −3.072, *p* = 0.002, respectively). Moreover, PSQI scores at the final follow-up were improved (t = 4.389, *p* < 0.001), and ESS scores at both 1 year post operation and the final follow-up were improved (t = 8.398, *p* < 0.001 and t = −2.598, *p* = 0.016, respectively). In terms of cognition, MMSE scores at 1 year post operation and the last follow-up showed no significant differences from baseline (Z = 2.070, *p* = 0.038 and Z = 0.144, *p* = 0.885, respectively). Moreover, compared with the baseline MOCA score of 27, scores were 28 at 1 year post operation (a 3.7% improvement; Z = 3.153, *p* = 0.002) and 27 at the final follow-up (Z = 2.646, *p* = 0.008). In terms of emotional function, the HAMD and HAMA scores at 1 year post operation were improved compared with the baseline (Z = −4.216, *p* < 0.001; t = 8.654, *p* < 0.001); there was no significant difference at the final follow-up. The average LEDD was reduced from 650 mg before surgery to 280 mg and 325 mg at 1 year post operation and the final follow-up, respectively (X^2^ = 4.094, *p* < 0.001 and X^2^ = 3.926, *p* < 0.001, respectively). This corresponds to an improvement of 56.9% at 1 year and 50.0% at the final follow-up (Table 3).

### 2.3. Complications of DBS Surgery in Older (≥75 years old) PD Patients

Postoperative complications occurred in six patients (26.1%). One patient (4.3%; Patient 7) had cortical entry point hemorrhage; the CT scan showed left frontal hematoma at 1 day post operation (Figure 2). After conservative treatment, the hematoma was completely absorbed. One patient (4.3%; Patient 17) had delayed healing of the left temporo-occipital skin. The wire was exposed at 4 months post operation; the wound healed well after debridement and sutures. The number of patients with postoperative hallucinations increased from three (17.4%; Patients 1, 3, and 12) to five (21.7%; Patients 1, 3, 4, 11, and 12); all hallucination symptoms were satisfactorily controlled after 1 week of antiparkinsonian drug adjustment. There were no scalp or chest incision infections in the group. During the DBS programming stage, two patients (8.7%; Patients 15 and 20) developed short-term dyskinesia symptoms; after repeated adjustments of stimulation parameters and drugs, their dyskinesia symptoms disappeared.

## 3. Discussion

### 3.1. Motor Symptom Changes after DBS

Age is an important element when considering DBS surgery. Charles et al. [34] reported the details of 56 consecutive PD patients with a mean age of 56.0 ± 7.7 years who received bilateral STN-DBS; age was negatively correlated with stimulation-related postoperative improvement. Similarly, Russmann et al. [35] reported that, of 52 PD patients (13 patients >70 years old; mean age 74 years) who underwent bilateral STN-DBS implantation, all UPDRS motor scores in the off-medication state improved (by around 22% at 48 months post operation) but there was less improvement in patients over 70 years of age than in the younger patients. In contrast, Derost et al. [36] compared the clinical effects of STN-DBS between PD patients <65 years old (n = 34, mean age 57.4 ± 4.9 years) and ≥65 years old (n = 53, mean age 68.8 ± 2.8 years) up to 2 years post operation, and reported no significant differences between the two groups in acute UPDRS-III motor scores. Fabienne et al. [37] reported similar findings; 45 patients (mean age 60 ± 9 years, range 40–73 years) with STN-DBS surgery were followed up for 24 months, and there was no significant correlation between age and UPDRS-III improvement. Given the inconsistent results of these studies, it remains unclear whether older adults should be actively treated with DBS. The potential risk–benefit ratio of DBS in older patients must, therefore, be carefully evaluated.

In the present study, the median UPDRS-III score was improved by 45.7% at 1 year and 8.6% at a mean duration of 55.08 months in older adults (mean age 77.0 years). These significant improvements in motor symptoms indicate that DBS in such patients has both short- and long-term positive effects. However, there are still very few studies that have reported clinical outcomes after DBS in patients >70 years of age.

A single-center study retrospectively assessed a prospective registry of 37 PD patients treated with DBS who were ≥70 years old (average age 72.45 years, range 70–81 years); patients experienced a 51% reduction in UPDRS-III scores (from 31.79 to 15.50) from baseline to an average of 42.2 months post operation [12]. Another study assessed the effects of age and disease duration on rigidity and dyskinesia scores after DBS. Mean dyskinesia scores had sustained reductions at 1 year post operation, and these improvements were significantly greater in patients ≥70 years old (n = 10, mean age 73.9 years) than in those <70 years old (*p* = 0.011; patients ≥70 years: 90% improvement, *p* = 0.016; patients <70 years: 53% improvement, *p* = 0.003), However, in rigidity scores, there were no significant changes at 1 year post operation [38]. Only one study has evaluated the efficacy of DBS in PD patients ≥75 years old. Sharma et al. [14]. reported that 30 older patients (mean age 77.5 years, range 75.0–84.5 years) had significant improvements in motor scores (27.3%) at a mean duration of 2.5 years post operation; these improvements were sustained for up to 4 years. Our results are consistent with those of previous investigations; for elderly patients (>70 years), the treatment effects of DBS can last for approximately 5 years. Our data also revealed that there are many common diseases (such as hypertension and diabetes) among the elderly; there is, therefore, a high rate of loss to follow-up. Moreover, with time, the treatment effects of DBS decline rapidly; UPDRS-III scores had a 45.7% improvement at 1 year but just an 8.6% improvement at final follow-up, at an average of around 4.5 years post operation. This may be because PD develops relatively rapidly in elderly patients; in our cohort, the median UPDRS-III score in the stimulation-off period decreased from −11.4% at 1 year to −28.6% at the last follow-up. Therefore, for the elderly (≥75 years old) PD patients, the short-term effect is certain, which can not only improve the motor symptoms, but also reduce the LEDD; in terms of long-term efficacy, although the statistics show that it is still effective 4.5 years post operation, the efficacy is moderate. The results of longer follow-up (such as 8 years and 10 years) are still unclear. A large sample and long-term follow-up are necessary.

### 3.2. LEDD Changes after DBS

In our group of patients, we observed not only postoperative motor symptom improvements, but also significant reductions in LEDD. The LEDD decreased from 650 mg before surgery to 280 mg and 325 mg at 1 year post operation and the final follow-up, respectively; this corresponds to reductions of 56.9% and 50.0%. Similarly, in a study by Fabienne et al. [37], of 45 patients (40–73 years old) who underwent STN-DBS, the LEDD was decreased by 59% at 24 months. Moreover, in a single-center study, 37 DBS-treated PD patients ≥70 years old had a 37% reduction in LEDD (preoperative 891.94 mg, postoperative 559.6 mg), as well as significant reductions in medication doses per day (preoperative 11.54, postoperative 7.97) at 42.2 months post operation [12]. In a study by Sharma et al. [14], of 30 older patients (range 75.0–84.5 years) who underwent STN-DBS, the mean LEDD was reduced from 1318.9 mg before surgery to 688.6 mg after a mean follow-up of 1 year and 602.5 mg after a mean follow-up of 2.5 years (i.e., a reduction of 47.8% at 1 year and 54.3% at 2.5 years). There was also a significant reduction in the bilateral STN-DBS group, whereas there were no significant postoperative differences in LEDD in the unilateral STN-DBS group [14]. Our results are consistent with those reported previously [12,14,37]. Together, these findings suggest that STN-DBS can reduce the LEDD in elderly patients, especially those taking large doses of levodopa with accompanying complications. Nonetheless, LEDD decreases are less significant in the elderly than they are in younger PD patients. For example, Russmann et al. [35] reported that, in 52 PD patients who underwent bilateral STN-DBS, patients >70 years old had 49% reductions in medication at the last follow-up, whereas younger patients had reductions of 74%. In the study by Derost et al. [36], similar LEDD reductions were observed in PD patients <65 years (49.5%) versus those ≥65 years (40.6%) at 3 months, which remained stable for up to 2 years. In the present study, we also found that the LEDD decreased significantly after surgery and remained low for a relatively long time.

### 3.3. Non-Motor Symptom Changes after DBS

QoL is correlated with mood, cognition, sleep, and other non-motor symptoms. Non-motor symptoms are positively correlated with age; older age corresponds to more severe non-motor symptoms, and these symptoms can seriously affect the QoL of elderly PD patients. DBS surgery has been reported to significantly improve non-motor symptoms [39,40,41,42,43,44] including PDQ-39, MMSE, HAMD, total PDSS, and total ESS scores, thus greatly improving the QoL of elderly PD patients [45]. A prospective open-label study conducted by the Non-Motor PD Study Group within the International Parkinson’s and Movement Disorder Society evaluated the effects of STN-DBS in 120 patients stratified by age (≤59, 60–69, and ≥70 years) with comparable disease duration and disease severity at baseline [46]. The authors reported that, despite a significant improvement in QoL (measured using the PDQ-8) in all age groups, the ≥70-year-old group had a lower effect size (0.42) than the ≤59 and 60–69 year old groups (0.83 and 0.59, respectively) [46]. The same reduced effect size with older age was also seen for ADL [46]. In contrast, in a retrospective study of the response to STN-DBS for up to 4 years in 30 patients with an average age of 77.5 years and mean disease duration of 11.8 years, no improvement in QoL was reported [14]. In another study, although there was no significant correlation between PDQ-39 subscales and age within the first year after DBS surgery, a significant negative correlation was observed at 24 months post operation (i.e., older PD patients had smaller improvements in QoL) [37]. In the current investigation, both the PDQ-39 and the ADL-Barthel Index score showed significant improvements: the PDQ-39 was improved by 37.5% and 8.0% at 1 year post operation and the final follow-up, respectively, and the ADL-Barthel Index was improved by 66.7% and 28.6% at the same timepoints. With increased postoperative time, there was a significant downward trend in the stimulation-off period; scores dropped to −40% of those of baseline. These findings indicate that QoL is improved after DBS for a short time in elderly PD patients, but improvements are not as sustained as those in young PD patients.

Previous studies have reported limited strict exclusion and inclusion criteria for DBS in elderly PD patients in terms of anxiety and depression assessments. However, considering the high incidence of anxiety and depression in elderly PD patients, as well as the psychological side-effects of antiparkinsonian drugs and psychosis after surgery, we strictly limited psychiatric symptoms in the present study. We excluded patients with moderate and severe depression (HAMD ≥ 17) and/or severe anxiety (HAMA ≥ 24). Furthermore, our results revealed that HAMD and HAMA scores were improved at 1 year post operation; however, there were no significant differences at the final follow-up. These findings are consistent with those of a meta-analysis that assessed anxiety and depressive symptoms in PD patients who underwent bilateral DBS; improvements in depression and anxiety were apparent after DBS and were more pronounced in the short term, whereas the effects seemed to wane In later assessments [47]. DBS therapy at either site appears to improve objective and subjective sleep parameters in patients with PD. This is most likely because of improvements in motor and some non-motor nocturnal symptoms; both increased total sleep time (by up to 1 h) and reduced sleep fragmentation have been noted [42]. Our data indicate that both short- and long-term sleep is significantly improved after DBS. However, the specific mechanism of improvement remains unclear, and it is unknown whether rapid eye movement behavior disorder is affected; further studies are needed.

One important concern when selecting candidates for DBS is the cognitive profile of patients. This is particularly true when selecting older patients because age is a predictor of cognitive decline after STN-DBS [48]. The results of a clinical trial evaluating the neuropsychological impacts of STN- and gPi-DBS in PD suggested that STN-DBS has a greater negative impact in cognitive tests after a 12 month follow-up; however, age and semantic verbal fluency at baseline are the only predictors of cognitive decline [49]. Because cognitive function declines rapidly in individuals aged ≥75 years old, the patients we selected for DBS surgery needed to have a normal cognitive function (MMSE ≥ 27; MoCA ≥ 26). The only exception was Patient 22 (83 years old), who had a MoCA score of 25; this patient had a strong demand for motor symptom improvement and accepted the risk of postoperative cognitive aggravation. After a reevaluation by the preoperative evaluation team, the patient was allowed to undergo surgery. In our patient group, cognitive function slightly improved following DBS implantation. Comparable findings were reported in a long-term follow up study; age at surgery was not correlated with the rate of cognitive decline after STN-DBS [46]. Similarly, in a meta-analysis of the neuropsychological results of 28 cohorts (612 patients) after STN-DBS, there was no association between postoperative changes in verbal fluency and patient age, disease duration, stimulation parameters, or change in dopaminergic dose after surgery [50]. These results clearly indicate the need for more predictive studies and suggest that a very careful assessment of patients’ cognitive reserve is currently necessary when considering DBS in the elderly.

### 3.4. Complications

The incidence of PD is higher in older individuals, and it is speculated that the disease may be related to advanced age. Elderly patients have poor tolerance, and PD in such patients is often accompanied by underlying cardiopulmonary diseases, such as hypertension and diabetes. Moreover, the risk of surgery is higher in the elderly than in younger people. Fabienne et al. reported significant age-related differences in the occurrence of cerebral bleeding; four patients (8.9%) had symptomatic cerebral bleeding, two of which (71 and 69 years old) died while the other two (70 and 71 years old) had transient neurological symptoms with full recovery [37]. However, in a retrospective analysis of 1757 DBS patients, older patients with PD (>75 years; 7.3% of the cohort) who were selected to undergo DBS surgery had a similar 90 day complication risk (including postoperative hemorrhage or infection) to their younger counterparts. The authors suggested that age alone should not be a primary exclusion factor for determining candidacy for DBS [51]. Similarly, Wakim et al. [52] reported perioperative complication rates after DBS surgery and found that patients ≥75 years old did not have significantly different rates of seizure, cerebrovascular accident, readmission within 90 days of discharge, explantation due to infection, or surgical revision compared with patients <75 years old. Although the risk of postoperative intracranial bleeding seemed higher in the older group (6.1%) than in the younger group (3.1%), this difference was not significant (*p* = 0.06). General postoperative complications in the present study occurred in 26.1% of patients, and they included cortical hemorrhage, delayed incision healing, hallucinations, and stimulus-related dyskinesia; there were no irreversible neurological deficits. Together, these findings indicate that older age should not be considered a contraindication to DBS surgery in PD patients who undergo a comprehensive evaluation and meet DBS surgical criteria.

The strengths and limitations of this study are as follows: there are few articles like ours that analyzed the efficacy and safety of DBS in elderly (≥75 years old) PD patients. In our article, not only were short-term and long-term motor symptoms, LEED, and aspects of non-motor symptoms evaluated, but so was the safety, providing some reference for readers engaged in functional neurosurgery. The limitations of this article were its single-center nature, retrospective analysis with a small sample size, a higher lost follow-up rate, the lack of a control group, a sham procedure by switching on or off the device post operation, and no analysis of the DBS prognostic factors in elderly patients.

#### Future Work

PD patients with a history of more than 4 years and an age of more than 75 years account for a considerable number. There has been a dispute about whether such elderly patients should be operated on or not. The final follow-up time was an average of 55.08 months in our group. The efficacy of DBS surgery in elderly PD patients still exists; however, 10 years after DBS surgery, the efficacy of DBS is still unknown. There are still many problems to be solved in DBS surgery for elderly (≥75 years old) PD patients. In the future, a multicenter, double-blind, prospective study is needed to determine whether elderly (≥75 years old) PD patients can benefit from DBS surgery, to study the factors related to the prognosis of DBS, as well as the impact of DBS on the life, and to provide evidence-based medical evidence for DBS surgery in elderly PD patients.

## 4. Conclusions

DBS showed efficacy and safety in older (≥75 years old) PD patients. DBS led to long-term postoperative improvements in motor symptoms, QoL, ADL, LEDD, and sleep, as well as short-term improvements in emotional and cognitive function. Elderly PD patients can, therefore, benefit from DBS if they meet the surgical indications.

## Figures and Tables

**Figure 1 brainsci-12-01588-f001:**
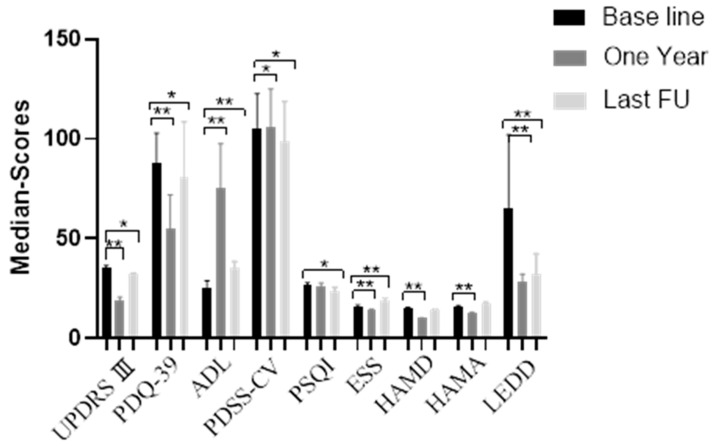
Visual analysis of histograms. * *p* < 0.05; ** *p* < 0.01.

**Figure 2 brainsci-12-01588-f002:**
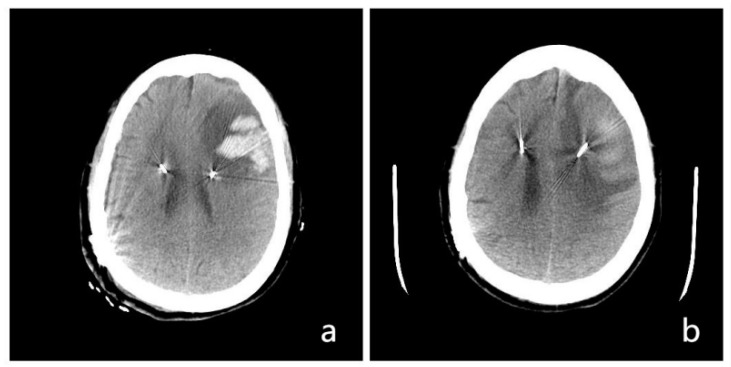
DBS operation with intracranial hemorrhage: (**a**) left frontal hematoma at 1 day post operation; (**b**) the hematoma was mostly absorbed at 21 days post operation.

**Table 1 brainsci-12-01588-t001:** Clinical characteristics of the 27 elderly PD patients (≥75 years old) treated with DBS surgery.

No.	Sex	Age (Y)	DOS (M)	Basic Disease and Surgical History	DBS Target	Lead Model	FU (M)	Complications
1	M	76	63	—	B-STN	PINSL301	83	—
2	F	75	180	Cholecystectomy	B-STN	PINSL301	52	—
3	M	76	111	Thyroidectomy;	B-STN	PINSL301	36	—
4	M	76	180	—	B-STN	Medt3389	30	Hallucinations
5	F	78	123	Thyroidectomy; cholecystectomy	B-STN	PINSL301	36	—
6	F	75	60	—	B-STN	PINSL301	30	—
7	F	75	192	Hypertension	B-STN	PINSL301	30	Left frontal hematoma
8	M	77	96	Diabetes; cholecystectomy	B-STN	PINSL301	33	—
9	M	76	72	—	B-STN	PINSL301	21	—
10	F	76	120	Thyroidectomy	B-STN	Medt3389	108	—
11	F	77	72	Hypertension; left pallidotomy	R-Gpi	Medt3387	94	Hallucinations
12	M	75	60	Hypertension; diabetes	B-STN	Medt3389	69	—
13	M	75	72	Hypertension	B-STN	Medt3389	65	—
14	M	75	48	Diabetes	B-STN	Medt3389	65	—
15	M	75	96	Thyroidectomy	B-STN	Medt3389	21	Dyskinesia
16	M	81	120	—	B-ViM	Medt3387	105	—
17	M	75	96	—	L-ViM, R-STN	Medt3389	41	Scalp incision delayed healing
18	M	80	432	—	B-STN	Medt3389	41	—
19	M	77	121	Hypertension; cholecystectomy	B-STN	PINSL301	31	—
20	F	75	72	—	B-STN	Medt3389	72	Dyskinesia
21	M	77	144	—	B-STN	PINSL301	66	—
22	M	83	60	Hypertension	B-Gpi	Medt3387	60	—
23	M	85	180	Hypertension	B-STN	Medt3389	78	—
24 *	M	76	62	—	B-STN	Medt3389	—	—
25 *	M	75	276	—	B-STN	Medt3389	—	—
26 *	M	86	444	Diabetes; resection of rectal cancer	B-Gpi	Medt3387	—	—
27 *	M	83	84	—	B-STN	Medt3389	—	—

PD: Parkinson’s disease; M: male; F: female; DBS: deep-brain stimulation; Y: year; M: month; DOS: duration of symptoms; B: both; L: left, R: right; ViM: thalamic ventral intermediate nucleus; STN: subthalamic nucleus; GPi: globus pallidus internus; Medt 3389/3387: Medtronic 3389/3387 electrodes (Medtronic, Ltd., Minneapolis, MN, USA); PINSL301: PINSL301 electrodes (Beijing PINS Medical Co., Beijing, China); FU: follow-up; “—”: none. * Patient lost to follow up.

**Table 2 brainsci-12-01588-t002:** UPDRS-III, PDQ-39, and ADL-Barthel Index scores at baseline, 1 year post operation, and last follow-up (on-/off-stimulation) in older (≥75 years old) PD patients.

Evaluation Scale	UPDRS III	PDQ-39	ADL-Barthel
Range of score	0–108	0–100	0–100
Baseline	35(4)	88(17)	25(15)
1 Year			
(Stimulation-on)	19(8) *	55(31) *	75(30) *
Z/t	19.230	11.390	−4.205
*p*	<0.001	<0.001	<0.001
(Stimulation-off)	39(10)	85(13)	15(15)
Z/t	3.456	−1.287	−3.225
*p*	0.002	0.198	0.01
last FU			
(Stimulation-on)	32(2) *	81(34) *	35(10) *
Z/t	−4.030	2.142	−4.034
*p*	<0.001	0.044	<0.001
(Stimulation-off)	45(4)	99(39)	15(5)
Z/t	−4.207	−2.801	−3.959
*p*	<0.001	0.005	<0.001

* *p* < 0.05 compared with preoperative baseline. UPDRS III: Unified Parkinson’s Disease Rating Scale Part 3; PDQ-39: 39-item Parkinson’s Disease Questionnaire; ADL-Barthel: Barthel Index for Activities of Daily Living; *p*: *p*-value; FU: follow-up.

**Table 3 brainsci-12-01588-t003:** Changes in sleep, cognition, and mood scores and LEDD in older (≥75 years old) PD patients at baseline, 1 year post operation, and final follow-up.

Evaluation Scale	Sleep			Cognition		Emotion		LEDD (mg)
	PDSS-CV	PSQI	ESS	MMSE	MoCA	HAMD	HAMA	LEDD
Range of score	0–150	0–42	0–24	0–30	0–30	0–76	0–56	
Baseline	105(17)	26.43 ± 5.91	16(5)	28(2)	27(1)	15(4)	15.91 ± 4.47	650(573)
1 Year	106(18) *	26.04 ± 6.56	14(4) *	28(1)	28(0) *	10(2) *	12.43 ± 4.17 *	280(150) *
Z/t	3.669	0.883	8.398	2.07	3.153	4.216	8.654	4.094
*P*	<0.001	0.387	<0.001	0.038	0.002	<0.001	<0.001	<0.001
last FU	99(20) *	23.74 ± 7.58 *	19(6) *	28(1)	27(1) *	14(5)	17.13 ± 6.45	325(300) *
Z/t	3.072	4.389	19.23	0.144	2.646	0.884	1.494	3.926
*p*	0.002	<0.001	<0.001	0.885	0.008	0.377	0.149	<0.001

* *p* < 0.05 compared with preoperative baseline. PDSS-CV: Parkinson’s Disease Sleep Assessment Scale—Chinese Version; PSQI: Pittsburgh Sleep Quality Index Scale; ESS: Epworth Sleepiness Scale; MMSE: Mini-Mental State Examination; MoCA: Montreal Cognitive Assessment; HAMD: Hamilton Depression Scale; HAMA: Hamilton Anxiety Scale; LEDD: levodopa equivalent daily dose; *p*: *p*-value; FU: follow-up.

## Data Availability

Not applicable.

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
