# Peer review of "Short- and Long-Term Efficacy and Safety of Deep-Brain Stimulation in Parkinson’s Disease Patients aged 75 Years and Older"

_brainsci, 2022, doi:10.3390/brainsci12111588_

Round 1

Reviewer 1 Report

Dear Authors; I found this study an interesting  investigation on the efficacy and safety of deep brain stimulation (DBS) in the treatment of patients with Parkinson’s disease aged 75 years and older. It needs some extra work to improve its presentation and clarity prior to further  processing it. Regards. P.S. 

[1] Writing:

[1-1]  Add list of used abbreviations at the end right before the reference section for reader easy access. 

[1-2]  References: Make sure to be in MDPI format. For example years for articles are in bold format. Also reference numbers are typed twice in them. Remove the double one.

[1-3]  In statistical analysis section highlight the "Improvement Rate(%)" formula in the middle by separating it from the text. Add an equation number to it. 

[1-4]  Add subsection "4.5. Future Work" to your discussion section and recommend few potential directions for further research to the readers. 

[2] Statistical:

[2-1]  Low Sample Size:  Only 27 participants is problematic.  Report the power of the tests performed in the study (need all to be minimum 80% to justify the results). 

[2-2]  Missing Table.1. Baseline Descriptive Statistics: Add this Table in section "3.1. Descriptive Statistics" and move up other Results subsection numbers for one unit:  3.1--> 3.2;  3.2-->3.3 so on.

[2-3]  Missing Visualization Figures: Add a  summary bar chart figure with key results of interest in the end of "3.1. Postoperative efficacy of DBS in elderly PD patients" to highlight the key results of the study. You bars will have 95% CI on their top as example below. Comment on the Figure. See the following chart for an example(last plot):

https://statistics.laerd.com/spss-tutorials/bar-chart-using-spss-statistics-2.php

Reviewer 2 Report

this is a very interesting paper for the dbs -community.

the conclusion is well supported and statistically well done.

nevertheless a control group of not operated pd-patients is missing to compare the real benefit of the intervention.

the results should be also controlled by a sham-procedure by switching on or of the device postop.

at least these two point should be discussed or noted in the limitations.

Reviewer 3 Report

An interesting work. However, it could be improved:

1) The effect of DBS con dyskinesia could have been assessed. Both subthalamic (see an cite PMID: 10201440) and pallidal stimulation could improve dyskinesia. 

2) The long-term improvement is somewhat modest. Despite is statistically significant, the clinical relevance is minimal. This should be emphasized by the authors. 

3) A discussion on the strengths and limitations of this study should be added.

Round 2

Reviewer 1 Report

Dear Authors; my main concerns were addressed satisfactorily. Regards.